# The microeconomics of abortion: A scoping review and analysis of the economic consequences for abortion care-seekers

Ernestina Coast[1]*, Samantha R. Lattof[1], Yana van der Meulen Rodgers[2,3], Brittany Moore[4], Cheri Poss[4]

1 Department of International Development, London School of Economics and Political Science, London, United Kingdom, 2 Department of Labor Studies and Employment Relations, Rutgers University, Piscataway, New Jersey, United States of America, 3 Department of Women's and Gender Studies, Rutgers University, Piscataway, New Jersey, United States of America, 4 Ipas, Chapel Hill, North Carolina, United States of America

* e.coast@lse.ac.uk

## Abstract

### Background

The economic consequences of abortion care and abortion policies for individuals occur directly and indirectly. We lack synthesis of the economic costs, impacts, benefit or value of abortion care at the micro-level (i.e., individuals and households). This scoping review examines the microeconomic costs, benefits and consequences of abortion care and policies.

### Methods and findings

Searches were conducted in eight electronic databases and applied inclusion/exclusion criteria using the PRISMA extension for Scoping Reviews. For inclusion, studies must have examined at least one of the following outcomes: costs, impacts, benefits, and value of abortion care or abortion policies. Quantitative and qualitative data were extracted for descriptive statistics and thematic analysis. Of the 230 included microeconomic studies, costs are the most frequently reported microeconomic outcome (n = 180), followed by impacts (n = 84), benefits (n = 39), and values (n = 26). Individual-level costs of abortion-related care have implications for the timing and type of care sought, globally. In contexts requiring multiple referrals or follow-up visits, these costs are multiplied. The ways in which people pay for abortion-related costs are diverse. The intersection between micro-level costs and delay(s) to abortion-related care is substantial. Individuals forego other costs and expenditures, or are pushed further into debt and/or poverty, in order to fund abortion-related care. The evidence base on the economic impacts of policy or law change is from high-income countries, dominated by studies from the United States.

### Conclusions

Delays underpinned by economic factors can thwart care-seeking, affect the type of care sought, and impact the gestational age at which care is sought or reached. The evidence

**Data Availability Statement:** All relevant data are within the paper and its Supporting information files.

**Funding:** This work was supported by the Netherlands Ministry of Foreign Affairs, activity number 28438. This funder had no role in the design and development of the manuscript or the decision to publish.

**Competing interests:** The authors have declared that no competing interests exist.

base includes little evidence on the micro-level costs for adolescents. Specific sub-groups of abortion care-seekers (transgendered and/or disabled people) are absent from the evidence and it is likely that they may experience higher direct and indirect costs because they may experience greater barriers to abortion care.

## Introduction

The socio-economic determinants of health and the inequitable distribution of the economic costs and consequences of healthcare are well established. For one type of health care—abortion-related care—the individual-level economic costs and consequences are concentrated among pregnant individuals. The global evidence about the individual-level economics of seeking and procuring abortion-related care has not been gathered and synthesised.

Understanding the microeconomics of abortion-related care allows us to pose, and consider how we might answer, some critical questions, including but not limited to: What does it cost individuals to procure abortion-related care, and do relative costs impact on decision-making about type and timing of care? How and in what ways do the economic consequences of care, or their anticipation, influence the timing and type of care sought, and its longer-term consequences? What economic value or benefit, if any, is attached to abortion-related care?

These questions, and their answers, matter from multiple linked framings of abortion-related care. Framing abortion-related care only as a public health issue—the health consequences of unsafe abortion are profound—means potentially missing critical lenses with important consequences. Issues of in/equity mean that we need to understand how the microeconomics of abortion are distributed within and across populations. Are poorer people more likely to seek less safe abortion because the costs of safer abortion care are beyond their reach? We know that the distribution of economic power is a critical social determinant of health. A reproductive justice perspective, moving beyond the enacting of reproductive rights, can improve the analytic understanding of how abortion-related care intersects with microeconomics. For example, if one effect of abortion criminalization is the higher likelihood of exclusion from safe abortion services of those who are unable to afford them, what are the consequences of differential resources on people's ability to seek or access abortion-related care?

This systematic mapping of the evidence on the microeconomics of abortion-related care uses four key economic components: costs, impacts, benefits, and values. The framework was developed by the authors to reflect our focus on the economics of abortion, rather than just the finances of abortion. As economics (like sociology) focuses on behaviours as well as money, the goal of our framework is to include outcomes—negative or positive—that go beyond financial outcomes as measured in monetary terms. Economic costs of abortion-related care are the amount paid to obtain abortion care; they do not start at point of treatment and are incurred directly and indirectly throughout the care-seeking trajectory (such as transport, food, accommodation). Access to financial resources, frequently linked to social support, may be critical to a person's ability to obtain abortion information and services. A pregnancy has short- and long-term direct and indirect costs for individuals. Economic impacts are the economic effect or influence of abortion-related care or policies. Examples include the extent to which the actual or perceived costs of abortion-related care might impact on the type of care sought; and the ways in which abortion policies or laws might lead to changes in the pricing of abortion-related care. Economic benefits are the advantages or profits gained from receiving abortion

care or from the implementation of abortion policies. Economic value refers to the importance, worth, welfare gains, or utility from receiving abortion care or the implementation of abortion policies. For example, individuals may value aspects of different types of safe abortion care.

By systematically scoping the global evidence for the first time across these four economic domains, this article establishes the substantive understandings and methodological approaches that have been used to understand the microeconomics of abortion-related care. Mesoeconomic and macroeconomic findings are reported elsewhere as are the links between the economics of abortion and stigma [1–3].

## Methods

We conducted a transparent and reproducible scoping review using the Preferred Reporting Items for Systematic reviews and Meta-Analyses extension for Scoping Reviews (PRISMA-ScR) tool and reporting guidelines for protocols [4, 5] (S4 Appendix). We did a scoping review rather than a systematic review because we wanted to uncover what is known about the microeconomic consequences of abortion care and abortion policies and anticipated that varied types of evidence would be found. Our scoping review is focused on all abortion-related care, irrespective of its effectiveness and safety. We are centering what people do with respect to seeking abortion-related care, including ineffective actions undertaken to induce an abortion.

The searches, application of in/exclusion criteria, screening and data extraction were conducted using rigorous protocol and data extraction tools [available online—6] for the PICOTS (Patient population, Intervention, Comparator, Outcome, Timing, and Setting) criteria (Table 1). Studies published in peer-reviewed journals on induced abortion and/or post-abortion care (PAC) in any world region were considered, provided that they reported qualitative and/or quantitative data on one of the following microeconomic outcomes of abortion care or abortion policies: costs, impacts, benefits, and value of abortion care or abortion policies.

Eight electronic databases were searched using combinations of relevant search terms (Table 2) adapted to the particulars of each electronic database [6]. We supplemented these searches with expert-recommended articles. We included items in English, French, Spanish, German and Dutch. We conducted the searches and application of inclusion/exclusion criteria according to the PRISMA-ScR flow approach [5]. No assessments of item quality were made, as the purpose of this scoping review is to describe and synthesize the extent of evidence. Therefore, as a scoping review that explicitly excludes a quality assessment of included studies, we do not "weigh" the evidence presented by authors in an included item. Where authors of an included study inferred an economic outcome [cost, impact, benefit, or value] on the basis

**Table 1. PICOTS criteria used in the scoping review.**

| PICOTS | |
|---|---|
| Populations | Individuals who obtained abortions or post-abortion care and members of their households |
| Interventions | Induced abortion (safe/unsafe), post-abortion care, and/or abortion policies |
| Control | None |
| Outcomes | Quantitative or qualitative data on:<br>• economic costs of abortion care or abortion policies<br>• economic impacts of abortion care or abortion policies<br>• economic benefits of abortion care or abortion policies<br>• - economic value of abortion care or abortion policies |
| Timeframe | 1 September 1994 to 15 January 2019 |
| Setting | Any |

**Table 2. Search terms and their combinations.**

| 1. Abortion terms | 2. Economic terms | 3. Impact terms |
|---|---|---|
| abort* | cost* | cost* |
| termination of pregnancy | econom* | benefit* |
| terminate pregnancy | price* | value* |
| pregnancy termination | financ* | impact* |
| pregnancy terminations | resource* | |
| postabortion | fee* | |
| post-abortion | tax* | |
| | expenditure* | |
| | GDP | |
| | gross domestic product | |
| | pay* | |
| | expens* | |

of their evidence, the findings in this manuscript explicitly state that this inference or hypothesis belongs to the author(s) of the included study.

We extracted data into Excel for five randomly selected studies in order to assure quality in data extraction. Following this check for quality assurance, we divided the remaining included studies for data extraction. As a scoping review, we did not assess the risk of bias of individual studies.

This analysis synthesizes the microeconomic evidence base and identifies evidence gaps on the costs and benefits of abortion to individuals seeking abortions and their households. We report the data using a systematic narrative synthesis in which the results are presented narratively and organized thematically, supplemented with tables of descriptive statistics on included studies and their outcomes.

## Results

### Descriptive statistics

Our search generated 19,653 items for screening (Fig 1). After duplicate removal, the 16,918 remaining items were title and abstract (TIAB) screened for inclusion. We determined eligibility of all items, and unclear items were discussed. Where exclusion could not be determined on TIAB, authors screened the full text. Decisions were made in favor of an inclusive approach where questions remained; 230 studies met all inclusion criteria.

Among the countries covered in the 230 studies on the microeconomics of abortion, more than a quarter of all the studies (64/230) focused exclusively on the United States of America (USA), and an additional five multi-country studies included the USA (Table 3). This dominance of the USA in studies of abortion reflects political attention, data availability, the institutional affiliation of authors, the location of funding and other resources for conducting studies, and our search strategy languages.

After the USA, the country with the most coverage in the final inventory of studies was India (n = 18). Similar numbers of studies have focused on countries in Africa (n = 45) and Asia (n = 40). Relatively few studies have focused on countries in Latin America and the Caribbean, and noticeably absent with just a few exceptions (including Egypt, Iran, and Israel) are studies in the Middle East and North Africa.

The majority of studies were quantitative, with 92 studies relying exclusively on quantitative methods and another 73 studies including both quantitative and qualitative methods

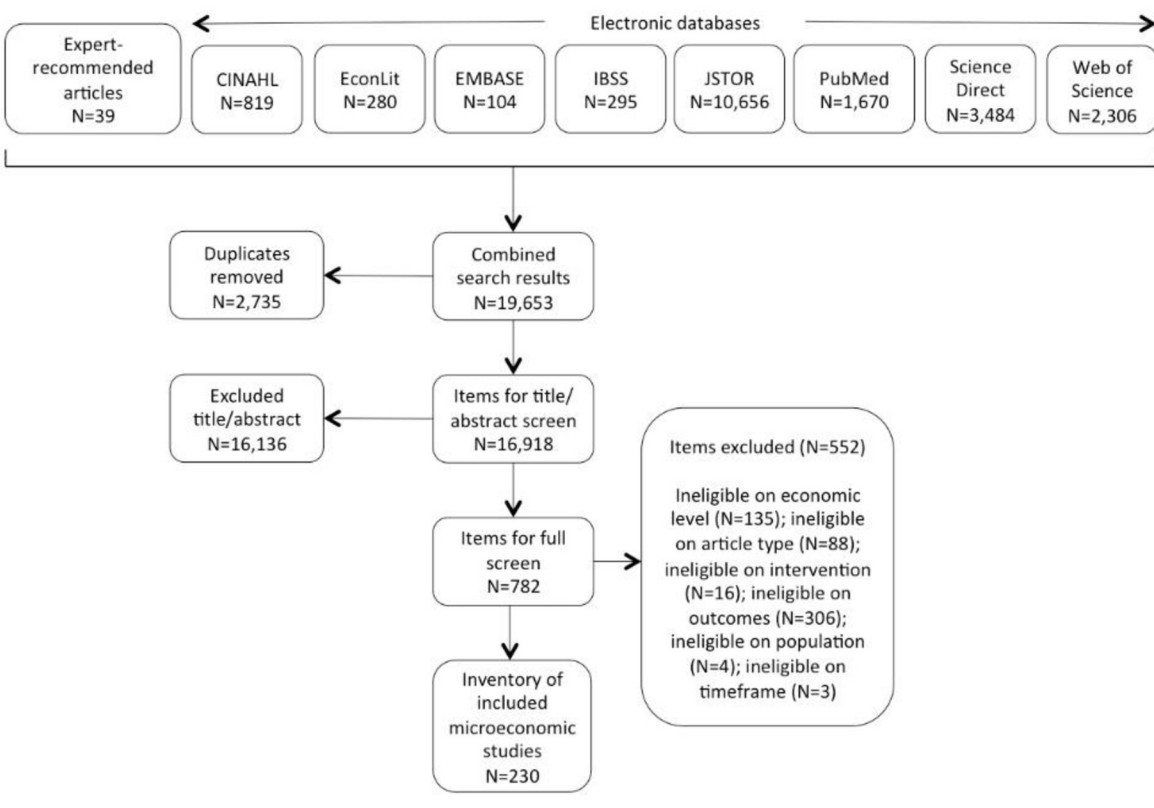

**Fig 1. Screening results.**

(Table 4). Nearly three-quarters of the lead authors were presumed to be women. Studies ranged in their level of geographic coverage, with the majority conducted at either a sub-national or health facility level. Study populations were most likely to be based on an individual's status as someone seeking abortion care.

We extracted data on the costs, impacts, benefits, and values of abortion services and abortion policies. Costs were reported most frequently (n = 180), followed by impact (n = 84), benefits (n = 39) and value (n = 26). To facilitate narrative analysis, we merged studies on benefits and values.

## Microeconomic costs

Microeconomic costs of abortion-related care (S1 Appendix) were the most frequently recorded outcome in our review. Many studies did not explicitly set out to study micro-level costs but include valuable evidence—quantitative and qualitative—underscoring the important role that economic costs play in trajectories to abortion-related care. The research design and level of detail relating to micro-level costs is heterogeneous, from cross-sectional direct costs of medical abortion drugs via telemedicine [7] to prospective direct and indirect costs of post-abortion compared to safe abortion care [8]. Seeking abortion-related care has—frequently substantial—costs for individuals, with potential implications for the timing and type of care sought, globally. Which costs are included, whether direct/indirect, is often unclear; in facility-based studies there tends to be a narrow focus on costs to, and at, the facility. Our review suggests that a much broader range of expenses and costs are important. We present

**Table 3. Included studies by region and country.**

| Region/country | # of studies | Region/country | # of studies |
|---|---|---|---|
| **Northern America** | **71** | **Europe** | **27** |
| United States | 64 | United Kingdom | 8 |
| Canada | 7 | Romania | 2 |
| | | Ireland | 3 |
| **Africa** | **45** | France | 1 |
| South Africa | 8 | Poland | 2 |
| Nigeria | 5 | Sweden | 2 |
| Ghana | 6 | Spain | 2 |
| Zambia | 4 | Netherlands | 1 |
| Kenya | 6 | Norway | 2 |
| Burkina Faso | 3 | Switzerland | 1 |
| Uganda | 3 | Turkey | 1 |
| Mozambique | 2 | Moldova | 1 |
| Ethiopia | 1 | Multiple countries | 1 |
| Cote d'Ivoire | 1 | | |
| Cameroon | 1 | **Latin America & Caribbean** | **20** |
| Egypt | 1 | Colombia | 1 |
| Multiple Countries | 4 | Mexico | 3 |
| | | Brazil | 3 |
| **Asia** | **36** | Chile | 3 |
| India | 16 | Guadeloupe | 2 |
| Thailand | 4 | Cuba | 1 |
| Bangladesh | 2 | Puerto Rico | 1 |
| Vietnam | 2 | Multiple countries | 6 |
| Nepal | 3 | | |
| Iran | 2 | **Oceana** | **9** |
| Indonesia | 1 | Australia | 8 |
| Pakistan | 1 | New Zealand | 1 |
| Cambodia | 1 | | |
| Myanmar | 1 | **Cross-Regional Studies** | **18** |
| Hong Kong | 1 | Global | 12 |
| Kazakhstan | 1 | Selected countries (including the US) | 5 |
| Israel | 1 | Selected countries (excluding the US) | 1 |
| | | **Total** | **230** |

the comparator costs when they are provided by the original study. Although rarely reported, respondents might not know the costs of their care [8, 9] or how costs were calculated [10].

The micro-level costs of abortion-related care are perhaps demonstrated most clearly in evidence from settings where abortion-related services are (theoretically) free-of-charge, including in Bangladesh [11], Canada [12] and South Africa [13], but where care involves direct or indirect costs. Limited evidence from India suggests that costs are linked to conditionality of care, with abortion services in the public sector provided for free only if the woman or her husband accepts some form of contraception, usually sterilization or an intrauterine device, post-abortion [14].

A subset of evidence considers individual difficulties to afford or pay costs of abortion-related care. In Kazakhstan, 40% of women identified 'financial problems' as the 'main difficulty' in obtaining an abortion [15]. Studies from the USA, reflecting the health insurance

**Table 4. Characteristics of included studies [n = 230].**

|  | *No. Studies* |
|---|---|
| **Type of Data** | |
| Quantitative | 92 |
| Qualitative | 65 |
| Both | 73 |
| **Methodology** | |
| Randomized controlled trial | 1 |
| Controlled clinical trial | 0 |
| Cohort analytic | 3 |
| Case-control | 0 |
| Cohort (before & after) | 4 |
| Interrupted time series | 0 |
| Qualitative | 64 |
| Mixed methods | 33 |
| Regression | 23 |
| Other | 81 |
| Review paper | 21 |
| **Inferred Gender of 1st Author** | |
| Woman | 169 |
| Man | 35 |
| Unclear | 26 |
| **Geographical Level** | |
| National | 49 |
| Sub-national (e.g. state, city) | 71 |
| Local (e.g. village) | 13 |
| Health facility | 67 |
| Other | 30 |
| **Study Population** | |
| Ethnic (or race) | 2 |
| National | 17 |
| Religion | 0 |
| Geographical location (e.g. urban/rural, region, facility) | 23 |
| Socio-economic | 1 |
| Age (e.g. adolescents) | 5 |
| Individual seeking an abortion | 70 |
| Multiple answers from list | 73 |
| Other | 28 |
| Abortion provider | 10 |
| Unclear / not specified | 1 |

context, find that it was somewhat or very difficult for 41% of respondents to pay for the procedure (52% among women not using health insurance) [16].

Some evidence considers the pregnancy outcomes for individuals unable to afford the costs of abortion-related care. In Thailand, three out of a sample of 30 women had abandoned attempts to obtain a termination because of the costs involved [17]. In Nepal, a landmark 2009 Supreme Court decision centered on a poor, rural woman who was forced to give birth to her sixth child due to her inability to afford the required fees for an abortion [18]. In the USA, among minors who identified as black or Hispanic, who received public health insurance

(Medicaid), or who had lower educational achievement, the risk of unintended birth was higher than among the general adolescent population. The authors suggest that it is the financial and time costs imposed by waiting periods with mandated multiple visits that may alter the pregnancy outcome [19].

**Indirect costs.**   The reasons for indirect costs of abortion are wide-ranging, including: companion costs [20]; childcare; overnight accommodation [21]; travel costs [22]; taking time off work; consumables (e.g. toiletries) [23]; and unofficial payments [24, 25]. In contexts where women have to seek multiple referrals or attend follow-up visits, these costs are multiplied [26]. Travel costs represented the most frequently cited indirect abortion-related costs at the micro level and in a wide range of contexts including in South Africa [27], the USA [28, 29] and Canada [30].

**Relative costs.**   Evidence that included relative comparator costs (e.g. average daily wage or monthly salary) was more meaningful than evidence that did not. Evidence from Poland showed that illegal abortions cost 2000–4000 PLN (US$ 500–1000), at a time when the average monthly Polish salary was 2000 PLN [31]. In Nepal, the basic fee (excluding other related costs) was Rs 645, representing approximately five days' wages for a female laborer [32]. Some authors reflect on how the relative costs of abortion-related care are likely to impact on the type of care sought. In Burkina Faso abortion costs ranged from a few thousand CFA francs for traditional abortifacients and up to 200,000 CFA francs for curettage in hygienic conditions. The monthly wage for a maid or caretaker is 20,000–40,000 CFA francs meaning that safer abortion methods were unaffordable for the poorest population groups [33]. In Kenya, where the cost of an abortion ranges from KS 60 for quinine purchased at a pharmacy to 5,000 KS (US$ 60) from a doctor, evidence suggests that even where women knew about a potentially safer option for abortion, the cost was prohibitive and limited them to less expensive options because most women earned less than 220 KS (US$ 2.50) per day [34].

**Resources for costs.**   People pay for abortion-related costs in diverse ways. In the USA, abortion care funds represent an important source for some women [16, 29]. In some contexts—linked to an inability to disclose their abortions, or an absence of other financial sources—women sought financing from credit/loans [35] and informal lenders [36]. In Kenya, patients unable to access a facility's required mode of payment (such as mobile money or the use of a credit/debit card) used brokers who charged a fee [23]. Women often had to borrow from their social networks (male partners, family, friends) in Nigeria [9], the USA [37, 38], Northern Ireland [36], Romania [39], Australia [20], Vietnam [40] and Brazil [41].

Men's roles in financing—knowingly or otherwise—abortion-related care are important across settings. In Zambia, few men who financially supported women seeking PAC were told the purpose of the care they were supporting [42]. In Australia, some women's ex-partners knowingly paid the abortion fee [20].

**Comparing costs of types of abortion-related care-seeking.**   Many of the research designs are comparative in nature, often comparing costs of various types of care seeking, including: different types of medical abortion drugs [7, 43]; medical abortion self-care compared to medical abortion formal care [44]; medical compared to surgical abortion [45–48]; safe abortion compared to PAC for induced abortion [8, 9, 24, 49]; and PAC for induced abortion compared to PAC for spontaneous abortion [50–52]. Two patterns emerge: micro-level costs for PAC for any induced abortion compared to care for a spontaneous abortion are substantially higher, often as a result of complex care-seeking trajectories due to the need for secrecy in restrictive settings; and, the costs of PAC for induced abortion are higher than those of safe abortion [24].

**Costs and delays.**   The intersection in the evidence between micro-level costs and delay(s) to abortion-related care, both induced abortion and PAC for induced abortion, is substantial.

In Zambia, the financial costs of seeking an abortion played a role in the timing and complexity of women's care-seeking trajectories, specifically finding money for transport [53]. Women who cannot access abortion in Northern Ireland must travel elsewhere to obtain one and pay as private patients; difficulties in obtaining funds can also lead to delays in obtaining an abortion, thereby increasing its cost [36]. In Kenya, a requirement that patients paid prior to each procedure restricted access to timely care, and inability to pay for services led to multiple referrals [23]. In the USA, health insurance processes and coverage played a role in influencing delays. In one study, although 59.6% of women had insurance, over half of participants paid out-of-pocket, and women with insurance reported complex processes and delays to obtain coverage [37]. In Colombia, women who were able to access a reproductive rights advocacy organisation were eventually able to obtain full insurance coverage, though they had their abortions later than they had desired [54].

**Costs and type of care sought.**   Actual and/or perceived costs of different types of abortion-related care impact care-seeking in a wide range of contexts. In Hong Kong, among adolescents and young women who had an illegal and unsafe abortion, the cost of safe abortion services was of concern to all the respondents [55]. In India, cost of services determined the choice of facility; women for whom cost was a concern sought care from those perceived to provide cheaper services, even when women had concerns about the provider's technical skills [56]. In Kenya, pregnancy termination in hospital settings and by 'high-profile' providers was considered very costly so women seek inexpensive but unsafe providers [57]. In Australia, women reported various reasons for not using surgical abortion services, despite the close proximity of services, including cost [58].

**Individual characteristics and costs of abortion-related care.**   The evidence base is heavily dominated by findings about adult women; we know less about costs as they pertain to adolescents [55, 59–61]. This is an important evidence gap because, globally, adolescents are less likely to be financially independent (or have ability to access sources of financing) compared to older women. Younger people may be charged higher rates for abortion-related services than older people. In India, because the law requires a guardian's consent for all medical care, including abortion services, for individuals aged below 18 years, girls reported that private practitioners were willing to forego this requirement in return for a fee up to five times the normal rate [62]. Evidence from the USA suggests that the costs of abortions were highest for very young adolescents (11–13 years); the authors suggest that this age group has 'significant difficulty' acquiring the funds needed for abortion procedures [63].

Age is just one factor, however, and individual characteristics can have substantial implications for abortion-related care costs. This diverse body of evidence underscores cross-cutting themes of in/equity, in/equality, in/justice and power in accessing and paying for abortion-related care. In some settings marital status is important: in India, unmarried women are vulnerable to unsafe abortions because of concerns about cost [64]. Migrant status impacts the costs and types of abortion-related care that women know about, seek and access [65], in countries as diverse as Guadeloupe [44], Great Britain [66], and Thailand [67].

Provider assessments of ability to pay are observed in many contexts, including Zambia [24], Chile [68, 69] and India [70]. However, differential provider pricing based on an ability to pay does not mean that wealthier women always pay more. In some settings it is the poorest (who may also be young, unmarried, undocumented, less educated) who pay the most for abortion-related care. In Burkina Faso, women from low-income households paid the highest amount for the abortion procedure and complications treatment [52]. Factors such as place of residence [31, 71, 72], occupational status [73], ethnicity [74], education [75, 76], and HIV status [17] also impact either the costs, or the ability to pay the costs, of abortion-related care. Evidence from the USA shows how state-level variation in changing Targeted Regulation of

Abortion Provider (TRAP) laws, the primary purpose of which is to limit abortion access, can have differential cost implications across multiple intersections [77].

**Interventions to assist with costs.** Interventions to assist with the costs of abortion-related care ranged from informal practices such as the provision of free services on a case-by-case basis to young Ghanaians who do not have money to pay [59], to informal community support systems in Myanmar for transport and a fund with donations for treatment of poor patients [78], to more formal clinic loan arrangements in Australia [21], to sliding fee scales in Mexico City [79], and American funds to assist women with abortion-related care costs [63].

## Microeconomic impacts

To describe the evidence base on the microeconomic impacts of abortion (S2 Appendix), we organize our analysis by research design. There are five main types of research design or evidence—with varying levels of inference. (i) Prospective research designs generating evidence from individuals at more than one point in time across their abortion trajectory. (ii) Analytic approaches to understand the impacts associated with policy and/or law changes. (iii) Studies that demonstrate (usually retrospective at the time of, or shortly afterwards, abortion) the economic impacts of abortion-related care costs. (iv) Studies comparing the economic impacts of different types of abortion care-seeking, including: medical compared to surgical abortion; PAC for induced abortion compared to spontaneous abortion; and, safe abortion compared to PAC for induced abortion. (v) Finally, studies that identify women's reasons for abortion (usually retrospective) can be used to infer the anticipated impacts of abortion, and give insights into the broad range of economic-related impacts that people anticipate as a result of having an abortion. Just one study explicitly sought to understand women's future aspirations as a result of having an abortion.

**Prospective studies.** There are a limited number of studies which use a prospective design; they demonstrate the potential power to understand how the overall costs of abortion-related care evolve over time. In Zambia, a prospective study showed how factors (age, wealth, education, marital status) intersected to influence not only how individuals financed their care, but also how this had implications for the type of care sought (safe abortion vs. PAC for induced abortion). Delays in fundraising increased both the cost and the risk of the procedure [8]. In Uganda, unsafe abortion resulted in deterioration in either the woman's or her family's economic circumstances, including: lost economic assets, incurred debt, lower consumption, increased work, or job loss [75].

**Analytic approaches to understand the impacts associated with policy and/or law changes.** These studies are predominantly USA-based [35, 38, 77, 80–83], with limited evidence from elsewhere [36, 84–86]. Qualitative evidence from low-income women who had abortions concludes that restrictive coverage policies appear to force women to take measures to raise money for an abortion that may have multiple consequences for health, wellbeing, and short and longer-term financial instability. These consequences then increase the difficulty of implementing an abortion decision. The authors identify 'ripple effects' for families of women seeking abortion services, and hypothesize that low-income abortion clients in states without public health insurance coverage of abortion experience more emotional and financial harm than clients in states where coverage is available [35].

A systematic review of USA TRAP laws found that these laws need not actually close clinics to have an impact. Laws that increase service costs or decrease availability of appointment slots could increase the time it takes to obtain an abortion. An increase in gestational age at presentation may limit the number of providers willing to perform an abortion (particularly if the pregnancy has entered the second trimester) and increase out-of-pocket costs to patients. The

authors hypothesize that while women with adequate resources are generally able to obtain an abortion with minimal difficulty, regardless of local policies, access-oriented barriers to abortion may introduce special challenges to low-income, young, and/or rural women who may be less able to manage increases in cost and distance [77].

A mixed-methods study of how women in the Republic of Ireland sought abortion services under conditions of restrictive laws concludes that these laws forced women into 'reproductive labor' [36]. A review of studies in Latin American found that requirements of prescriptions for medical abortion are a barrier that encourage use of informal, often more costly, sources of medical abortion [85]. In Benin, misoprostol was sought in pharmaceutical markets to avoid navigating the logistics of having to obtain a prescription [86].

**Studies on the economic impacts of abortion-related care costs.** These studies (usually retrospective at the time of, or shortly afterwards, abortion) identify a range of (often multiple) impacts in diverse contexts, including: education [24, 61, 87, 88]; employment/work/income [24, 40, 89–91]; foregone expenditures or increased debt and/or poverty [52, 92–94]; and, costs to mental health [16].

**Studies comparing the economic impacts of different types of abortion care-seeking.** Studies in this category include: medical compared to surgical abortion [90, 91]; PAC for induced abortion compared to spontaneous abortion [52, 95]; and, safe abortion compared to PAC for induced abortion [8]. Evidence from African countries comparing the economic impact of post-abortion care for induced abortion with either spontaneous abortion [52, 95] or safe abortion [8, 24] shows in all cases higher microeconomic impacts for PAC compared to other care-seeking.

Four studies—all from the USA—identify the impact of economic costs of abortion-seeking on delays to care-seeking and continuing a pregnancy [35, 82, 96, 97]. Delays linked to difficulties in navigating insurance coverage, referral, securing costs were all implicated in these studies. Other studies have uncovered further evidence on how economic impacts are implicated in delays to abortion care-seeking [27, 92, 97–99]. In Australia, women who experienced difficulties in financing the abortion had significantly higher odds of presenting for care at later than 9 weeks gestation [92].

**Studies that explore reasons for abortion.** Usually retrospective, these studies can be used to infer the anticipated economic impacts of having an abortion. Reasons reported in the evidence from diverse contexts includes: education [40, 89, 100–102], employment/occupation [40, 89, 100], wealth/poverty [39], caring for dependents [103], current and future relationships [100], and wellbeing of pre-existing children [10, 100]. However, perceived economic impacts (and reasons for having abortion) are rarely singular and are frequently intertwined [88, 103–105].

Finally, one study explicitly sought to understand women's future aspirations as a result of having had an abortion [106]. The authors generated evidence on women's one-year plans post-abortion among four groups: First Trimesters (presented in the first trimester, received abortion), Near-Limits (presented up to 2 weeks under the limit, received abortion), Non-Parenting Turnaways (included Turnaways who subsequently had an abortion elsewhere, reported that they had miscarried, or placed the child for adoption) and Parenting Turnaways (women with children who presented up to 3 weeks over the facility's gestational age limit, were turned away). One-year plans were related to areas including education, employment and change in residence. First Trimesters and Near-Limits were over 6 times as likely as Parenting Turnaways to report aspirational one-year plans. Among all plans on which achievement was measurable, Near-Limits and Non-Parenting Turnaways were more likely to have both an aspirational plan and to have achieved it than Parenting Turnaways [106].

## Microeconomic benefits and values

Microeconomic benefits (advantages) and values (importance, worth, welfare gains, utility) include diverse factors of intrinsic worth at the individual level. Very little evidence specifically uses the language of the economic benefits and values of abortion; much of the evidence included here is based on interpretation of relevant evidence. We identify three groups of studies on the microeconomic benefits and values of abortion-related care (S3 Appendix): (i) the ways in which laws and policies, including financing and insurance, have benefits and values to women; (ii) the benefits and values of different types of abortion-related care; and (iii) evidence of benefits and values derived from women's reasons for having an abortion.

**Benefits and values of financing.** This evidence—specifically insurance—is USA-based and limited to two studies. Qualitative evidence from low-income women shows that when full coverage of abortion in Medicaid is available, there is 'rarely a scramble for money that provokes feelings of indignity or delays abortion care' (p.1582) [35].

**Benefits and values of different types of abortion-related care.** This evidence either compares surgical and medical abortion [13, 58, 85, 107–109] or considers the perceived benefits of mHealth/telemedicine interventions [110, 111]. In the Republic of Ireland and Northern Ireland, a study of women who requested at-home medical abortion through online telemedicine suggests that women with few economic and social resources valued the lower costs of telemedicine compared to having to travel for an abortion [110].

**Benefits and values derived from women's reasons for having an abortion.** Evidence in this category relates to a range of factors, including: economic in/ability to afford or cope with a/nother child; pregnancy timing; costs of pregnancy/childbirth (distinct from costs of a child); partner and others' influences; positive implications for existing children; avoidance of health-related issues; avoiding pregnancy at a young age; continuation of education; and sex-selection.

There is substantial evidence globally about the economic benefits and values of avoiding having a/nother child [67, 93, 100, 102, 103, 112–115]. Particularly among adolescents and younger women, the ability to continue with or pursue education was an important benefit of abortion in diverse settings: USA [116], Ghana [100], Brazil [87], New Zealand [117], Guadeloupe [118], and India [62].

Relationship issues, whether to avoid violence in a controlling relationship in the United Kingdom (UK) [66], avoid becoming a second wife in Ghana [100], or to end a relationship due to pregnancy in Colombia [54], are layered into the benefits and values that women describe. For unmarried women, whether for issues of stigma in Indonesia [119] or because single motherhood was unaffordable in the USA [116], their marital status further added to the impacts of abortion.

Two studies articulated the benefits and values in terms of the positive implications for existing children of individuals who have an abortion in the Republic of Ireland and Northern Ireland [110] and India [62]. The value of sex selective abortion is inferred—not based on people's reported views—by authors of two studies from Canada [120] and France [121]. Many individuals may perceive or experience multiple intersecting and overlapping benefits and values from abortion [38, 94, 116, 122].

The benefits and values of identity maintenance as a result of abortion are also evidenced [119, 123]. In an Australian study that interviewed women seeking abortion services, the most important change reported by the women was an increased capacity to run their own lives. Women discovered that they could make and carry out difficult decisions, and that they could alter the course of events and exert their wishes over their destiny [123].

## Discussion

Although relatively few micro-level studies are defined explicitly by their authors or their methodology as "economic" studies, our review shows that there is a wealth of economically relevant information that can be gleaned from the evidence base. We draw out the substantive and methodological implications of our results for future research and identify some of the evidence gaps.

At the microeconomic level, the interplays between economics and delays to abortion-related care are striking. Across diverse contexts and populations, economic factors influence delays to decision-making about abortion-related care, attempts to seek care and the receipt of care. The Three Delays Model [124], which was originally developed for maternal healthcare seeking has been adapted and applied to abortion-related care [54, 125], offers a framework that could be more fully exploited in abortion-related care research. By unpacking the points at which economic factors introduce or compound delays to abortion-related care, greater insight into the points at which information and services might be better designed to reduce delays can be achieved. By further unpacking the intersections of these economic factors, we can better understand the ways in which health systems and contexts reproduce injustices and inequities. For example, it is often poorer individuals and/or adolescents who are least likely to be able to navigate or surmount economic barriers to abortion-related care.

Delays underpinned by economic factors can thwart care-seeking, affect the type of care sought, and impact the gestational age at which care is sought or reached. Although rarely explicitly included in evidence, the timing of confirmation of pregnancy is also likely to be strongly influenced by intersecting economic factors. We continue to know very little about the ways in which economic factors (including perceptions of economic factors) intersect with concepts of risk/safety and quality of care to affect abortion-related care-seeking and its timing. The limited evidence base suggests that the microeconomic costs of abortion impact on decision-making about the type of abortion care sought. In contexts where less safe abortion methods are cheaper than safer alternatives, there are profound implications for health outcomes.

Our scoping review identifies multiple gaps in our understanding of, and the evidence base for, the economics of abortion. The self-use/-management of medical abortion is tightly connected to the economics that surround it. Many gaps remain in our evidence base around the microeconomic impacts of abortion, including the indirect economic impact of abortion-related care and its longer-term economic impacts. We know very little about how the un/supportability or un/wantedness or un/plannedness or ambivalence around pregnancy intersects with economic benefits and values at the micro-level. We continue to know very little about the ways in which—conceptually separate from delays but linked in terms of health outcomes —economic factors intersect with concepts of abortion risk/safety and quality of care.

Methodologically, we know relatively little about the individual-level economic burden of seeking and procuring abortion. Particularly facility-based studies focus on treatment costs, however costs are incurred directly and indirectly throughout the treatment pathway (e.g. transport, food, accommodation, loss of income). The use of non-financial measures to assess the micro-level costs of abortion-related care is important, given that respondents might not know the costs of their care. The limited prospective evidence base suggests that there are likely substantial post-facility economic costs and impacts. Another knowledge gap is the extent— over time—of financial duress for abortion care-seekers and the people who support them [35]. In addition, very limited work has explored how women feel about having to obtain economic resources for abortion-related care from others.

There is great heterogeneity in what is in/excluded in understandings of individual costs and impacts in individual studies; the evidence base would benefit from a broader

understanding of in/direct costs—studies that focus solely on what abortion seekers pay underestimate the total costs (such as lost income or earnings and unofficial payments). The inadequacies of data on reported costs for abortion-related care are an important methodological issue that is rarely tackled or documented. The inclusion of comparators such as average monthly earning (not simply conversion of reported costs to US$) would benefit the evidence base, in order to situate the relative economic consequences of abortion-related care-seeking. Our review suggests that the field would benefit from greater harmonization of, and transparency about, the types of costs that are in/excluded. There are hints in the evidence that the changing architecture of financial systems will need to be accounted for in the evidence base; for example, the use of non-cash payment systems (debit/credit cards, mobile money) may lead to the increasing exclusion of individuals who operate in a cash-based economy.

Rarely are the conditional contexts of abortion-related care explicitly considered; limited evidence suggests that in some contexts the economic costs or impacts are linked to conditionality of care (such as acceptance of long-acting reversible contraception). Specific sub-groups of abortion care-seekers are not present in the evidence base—transgendered and/or disabled people—and given that they may experience greater barriers to abortion care, it is likely that they may experience higher direct and indirect costs. Finally, the microeconomics of second trimester abortion-related care are rarely explored, which is a substantial gap given the critical role of delays in abortion care-seeking.

The evidence base around the economic impacts of abortion policy or law change is entirely based on findings from high-income countries, dominated by studies in the USA. More generally, economic impact evidence from low- and middle-income countries is limited to findings from a few countries. Many gaps remain in our evidence base around the microeconomic impacts of abortion, including the indirect economic impact of abortion-related care and the longer-term economic impacts—both positive and negative—of abortion-related care.

Our scoping review is limited in its purview, excluding grey literature, published literature outside of peer-reviewed journals and relevant literature published in languages other than English, French, Spanish, German or Dutch. Nonetheless, it has highlighted many lacunae— geographically, substantively and methodologically. Our review underscores the critical importance of economic factors for people's abortion-related care-seeking. Whether framed by the social determinants of health, structural violence [126] or by an issue of reproductive justice, it is clear that if the economic dimension of abortion-related care-seeking is not taken into account by policies, laws, and health systems, then the outcomes will continue to be inequitable and unjust.

## Supporting information

**S1 Appendix. Summary of studies reporting microeconomic costs.**
(DOCX)

**S2 Appendix. Summary of studies reporting microeconomic impacts.**
(DOCX)

**S3 Appendix. Summary of studies reporting microeconomic benefits/value.**
(DOCX)

**S4 Appendix. Preferred reporting items of systematic reviews and meta-analyses extension for scoping reviews (PRISMA-ScR) checklist.**
(DOCX)

## Acknowledgments

We wish to thank Elaine Zundl, Lisbeth Gall, and Joe Strong for their assistance with screening and data extraction.

## Author Contributions

**Conceptualization:** Ernestina Coast, Samantha R. Lattof, Yana van der Meulen Rodgers, Brittany Moore, Cheri Poss.

**Data curation:** Ernestina Coast, Samantha R. Lattof, Yana van der Meulen Rodgers, Brittany Moore, Cheri Poss.

**Formal analysis:** Ernestina Coast, Samantha R. Lattof, Yana van der Meulen Rodgers, Brittany Moore, Cheri Poss.

**Investigation:** Ernestina Coast, Samantha R. Lattof, Yana van der Meulen Rodgers, Brittany Moore, Cheri Poss.

**Methodology:** Ernestina Coast, Samantha R. Lattof, Yana van der Meulen Rodgers, Brittany Moore, Cheri Poss.

**Project administration:** Ernestina Coast, Samantha R. Lattof, Yana van der Meulen Rodgers, Brittany Moore, Cheri Poss.

**Supervision:** Ernestina Coast, Samantha R. Lattof, Yana van der Meulen Rodgers, Brittany Moore, Cheri Poss.

**Validation:** Ernestina Coast, Samantha R. Lattof, Yana van der Meulen Rodgers, Brittany Moore, Cheri Poss.

**Writing – original draft:** Ernestina Coast, Samantha R. Lattof, Yana van der Meulen Rodgers, Brittany Moore, Cheri Poss.

**Writing – review & editing:** Ernestina Coast, Samantha R. Lattof, Yana van der Meulen Rodgers, Brittany Moore, Cheri Poss.

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
