## [Decision Letter · Decision Letter 0]

20 Jul 2020

PONE-D-20-01114

The microeconomics of abortion: A scoping review and analysis of the economic
consequences for abortion care-seekers

PLOS ONE

Dear Dr. Coast,

Thank you for submitting your manuscript to PLOS ONE. After careful consideration, we
feel that it has merit but does not fully meet PLOS ONE’s publication criteria as it
currently stands. Therefore, we invite you to submit a revised version of the
manuscript that addresses the points raised during the review process.

I acknowledge that Reviewer 2's suggestions are likely to be beyond the scope of this
paper. However, Reviewer 1 has provided a thorough review and raised a number of
pertinent points in the attached document that need to be addressed.

Please submit your revised manuscript by Aug 23 2020 11:59PM. If you will need more
time than this to complete your revisions, please reply to this message or contact
the journal office at plosone@plos.org. When
you're ready to submit your revision, log on to https://www.editorialmanager.com/pone/ and select the 'Submissions
Needing Revision' folder to locate your manuscript file.

If you would like to make changes to your financial disclosure, please include your
updated statement in your cover letter. Guidelines for resubmitting your figure
files are available below the reviewer comments at the end of this letter.

We look forward to receiving your revised manuscript.

Kind regards,

Hannah E Carter

Academic Editor

PLOS ONE

Journal Requirements:

"."

"-"

5. Please amend your manuscript to include your abstract after the title page.

Reviewers' comments:

Reviewer's Responses to Questions

**Comments to the Author**

1. Is the manuscript technically sound, and do the data support the conclusions?

Reviewer #1: Yes

Reviewer #2: No

2. Has the statistical analysis been performed
appropriately and rigorously? 

Reviewer #1: N/A

Reviewer #2: No

3. Have the authors made all data underlying the
findings in their manuscript fully available?

Reviewer #1: Yes

Reviewer #2: Yes

4. Is the manuscript presented in an intelligible
fashion and written in standard English?

Reviewer #1: Yes

Reviewer #2: Yes

5. Review Comments to the Author

Reviewer #1: Please see attached word document for the full details of my comments
and the breakdown associated with my completed review. I recommend Major
Revision

Many Thanks

Reviewer #2: I appreciate your study of the literature on abortion, which is helpful
to the extent that we can understand better what has been done. At the same time, it
would be desirable to conduct an independent research, based on your own research
design, in order to quantitatively find out what makes abortion difficult.

6. PLOS authors have the option to publish the peer
review history of their article (what does this mean?). If published, this will
include your full peer review and any attached files.

If you choose “no”, your identity will remain anonymous but your review may still be
made public.

**Do you want your identity to be public for this peer review?** For
information about this choice, including consent withdrawal, please see our
Privacy Policy.

Reviewer #1: No

Reviewer #2: No

---

## [Author Response · Author response to Decision Letter 0]

7 Dec 2020

Please see detailed attachment with responses

econ response to reviewers 271120.docx
---

## [Decision Letter · Decision Letter 1]

3 Mar 2021

PONE-D-20-01114R1

The microeconomics of abortion: A scoping review and analysis of the economic
consequences for abortion care-seekers

PLOS ONE

Dear Dr. Coast,

Thank you for submitting your manuscript to PLOS ONE. After careful consideration, we
feel that it has merit but does not fully meet PLOS ONE’s publication criteria as it
currently stands. Therefore, we invite you to submit a revised version of the
manuscript that addresses the points raised during the review process.

Please submit your revised manuscript by Apr 17 2021 11:59PM. If you will need more
time than this to complete your revisions, please reply to this message or contact
the journal office at plosone@plos.org. When
you're ready to submit your revision, log on to https://www.editorialmanager.com/pone/ and select the 'Submissions
Needing Revision' folder to locate your manuscript file.

If you would like to make changes to your financial disclosure, please include your
updated statement in your cover letter. Guidelines for resubmitting your figure
files are available below the reviewer comments at the end of this letter.

We look forward to receiving your revised manuscript.

Kind regards,

Hannah E Carter

Academic Editor

PLOS ONE

Journal Requirements:

Reviewers' comments:

Reviewer's Responses to Questions

**Comments to the Author**

1. If the authors have adequately addressed your comments raised in a previous round
of review and you feel that this manuscript is now acceptable for publication, you
may indicate that here to bypass the “Comments to the Author” section, enter your
conflict of interest statement in the “Confidential to Editor” section, and submit
your "Accept" recommendation.

Reviewer #1: (No Response)

Reviewer #2: All comments have been addressed

2. Is the manuscript technically sound, and do the data
support the conclusions?

Reviewer #1: Yes

Reviewer #2: Yes

3. Has the statistical analysis been performed
appropriately and rigorously? 

Reviewer #1: N/A

Reviewer #2: Yes

4. Have the authors made all data underlying the
findings in their manuscript fully available?

Reviewer #1: Yes

Reviewer #2: Yes

5. Is the manuscript presented in an intelligible
fashion and written in standard English?

Reviewer #1: Yes

Reviewer #2: Yes

6. Review Comments to the Author

Reviewer #1: The authors have significantly improved the paper; I’m very glad that
they incorporated the suggested edits I had made in my first review. All line
numbers below are from the clean transcript.

I do still find some of the evidence presented a bit thin, e.g. lines 292-294 &
lines 400-402, by which I mean one reads the text and still walks away not
understanding the what the results are of the included articles.

I will stand by what I said in the first review, that there’s too much in this one
manuscript, and I would specifically remove (or seriously winnow) the section on
“benefits or values” as a lot of the information described there does not fall into
the economic sphere, including relationship issues, stigma, and identity
maintenance. If the authors insist on keeping this in the paper, they should make
the link between these topics and microeconomics much clearer.

I think this conclusion is very important: “Although relatively few micro-level
studies are defined explicitly by their authors or their methodology as “economic”
studies, our review shows that there is a wealth of economically relevant
information that can be gleaned from the evidence base.” I also find valuable their
specification in research gaps presented in the Discussion.

Minor points:

They are still overusing the en-dash on pg. 2 and 12, and they’re still mixing
hyphens with en-dashes (lines 80-81, line 190, line 275, line 308, lines 331-332,
lines 481-482, lines 491-492).

Line 102: “(PRISMA)” acronym not necessary because it appears on line 103.

Line 136: why “simultaneously”?

Lines 156-159: Another reason for the dominance of studies from the USA is the
languages the authors used in which to conduct their search.

Line 170: “three-quarters”

Line 173: Still hard to know what is meant by “abortion care-seeker status”—this is a
term that the author team is clearly comfortable with but which is not intuitive to
the reader. How about, “Study populations were most often whoever was seeking an
abortion.” (I don’t even think they need the text “rather than…”)

Line 217: “the” missing before “pregnancy outcome.”

Lines 257-258: Stating that the collection of non-financial measures are important is
a recommendation, and not a result, and should be moved to the Discussion.

Line 270: insert “any” before “induced abortion”

Line 274: the word “intersection” is not the right word here; replace with
“relationship between”?

Line 283: the word “interactions” is not the right word here.

Line 287: what is “an advocacy service”?

Lines 299-230: “women reported various reasons for not using surgical abortion
services…because of the cost”?

Lines 306-307: “Younger ages may mean higher costs charged for abortion-related
services”—this is unnecessarily obtuse.

Line 335: Not sure that the word “interventions” is the correct one here; it sounds
as though the section would be talking about things like “The Justice Fund” in the
USA which was a time-delimited experiment which paid for period for women’s
abortions. How about “Social support” or “Charitable accommodations”?

Line 361: It’s not the fact that there are a limited number of studies which
demonstrates the power of prospective research designs, but rather “There are a
limited number of studies which use a prospective design; they demonstrate the
potential power to understand how the overall costs…”

Line 372 and the rest of the paragraph, and the paragraph which follows about TRAP
laws: although it is possible to cross-reference, the authors should specify in the
text geographically where the evidence is coming from.

Line 394: “prescriptions” for medication abortion? “medication” �medication
abortion?

Lines 400-402: the references are not formatted properly, and there’s a gratuitous
period and space after [100]

Line 450: “in” should be “on”

Line 501-502: That text belongs at the beginning of the section.

Line 539: Indent missing.

Reviewer #2: The revised version appears to take into account the comments, so I have
no other comments at this time.

7. PLOS authors have the option to publish the peer
review history of their article (what does this mean?). If published, this will
include your full peer review and any attached files.

If you choose “no”, your identity will remain anonymous but your review may still be
made public.

**Do you want your identity to be public for this peer review?** For
information about this choice, including consent withdrawal, please see our
Privacy Policy.

Reviewer #1: No

Reviewer #2: No

---

## [Author Response · Author response to Decision Letter 1]

17 Apr 2021

See attached file with detailed responses to reviewer.

econ RR2 response document 170421.docx
---

## [Editor Report · Decision Letter 2]

10 May 2021

The microeconomics of abortion: A scoping review and analysis of the economic
consequences for abortion care-seekers

PONE-D-20-01114R2

Dear Dr. Coast,

We’re pleased to inform you that your manuscript has been judged scientifically
suitable for publication and will be formally accepted for publication once it meets
all outstanding technical requirements.

Kind regards,

Hannah E Carter

Academic Editor

PLOS ONE
---

## [Editor Report · Acceptance letter]

31 May 2021

PONE-D-20-01114R2 

The microeconomics of abortion: A scoping review and analysis of the economic
consequences for abortion care-seekers 

Dear Dr. Coast:

I'm pleased to inform you that your manuscript has been deemed suitable for
publication in PLOS ONE. Congratulations! Your manuscript is now with our production
department. 

Kind regards, 

on behalf of

Dr. Hannah E Carter 

Academic Editor

PLOS ONE